# Study of Acute and Sub-Acute Effects of Auditory Training on the Central Auditory Processing in Older Adults with Hearing Loss—A Pilot Study

**DOI:** 10.3390/ijerph17144944

**Published:** 2020-07-09

**Authors:** Carla Matos Silva, Carolina Fernandes, Clara Rocha, Telmo Pereira

**Affiliations:** 1Departamento de Audiologia, Instituto Politécnico de Coimbra, ESTeSC-Coimbra Health School, 3046-854 Coimbra, Portugal; 2LABINSAÚDE-Laboratório de Investigação em Ciências Aplicadas à Saúde, Instituto Politécnico de Coimbra, ESTeSC, 3046-854 Coimbra, Portugal; telmo@estescoimbra.pt; 3School of Arts and Humanities, Centro de Linguística da Universidade de Lisboa, University of Lisbon, 1649-004 Lisboa, Portugal; 4MiniSom, uma marca amplifon, 1050-148 Lisboa, Portugal; carol_sofia_fernandes@hotmail.com; 5Departamento das Ciências Complementares, Instituto Politécnico de Coimbra, ESTeSC-Coimbra Health School, 3046-854 Coimbra, Portugal; ClaraPR@estescoimbra.pt; 6INESC Coimbra, Institute for Systems Engineering and Computers at Coimbra, University of Coimbra, 3000-033 Coimbra, Portugal; 7Departamento de Fisiologia Clínica, Politécnico de Coimbra, ESTeSC-Coimbra Health School, 3046-854 Coimbra, Portugal

**Keywords:** auditory training, auditory perception, elderly people, hearing loss

## Abstract

Background: Impairment in speech perception is a common feature of older adults. This study aimed at evaluating the acute and sub-acute (after three months) effects of auditory training on central auditory processing in older people with hearing loss. Methods: A nonrandomized study was conducted enrolling 15 older adults with hearing loss and an average age of 78.6 ± 10.9 years. All participants underwent a baseline otoscopy, tympanogram, audiogram and speech-in-noise test with a signal-noise ratio (SNR) of 10 and 15 dB. Afterwards, auditory training intervention was implemented consisting of 10 training sessions over 5 weeks. Participants were divided into two groups: group 1 (G1) underwent auditory training based on a speech-in-noise test; group 2 (G2) underwent a filtered-speech test. Auditory processing was evaluated at baseline (T0) immediately after the intervention (T1) and 3 months after the intervention (T2). Results: Group 1 were quite efficient regardless of the SNR in the right ear with statistically significant differences from T0 to T1 (*p* = 0.003 and *p* = 0.006 for 10 dB and 15 dB, respectively) and T0 to T2 (*p* = 0.011 and 0.015 for 10 dB and 15 dB, respectively). As for the left ear, the increase of success was statistically significant for the SNR of 10 dB and 15 dB from T0 to T1 (*p* = 0.001 and *p* = 0.014, respectively) and from T0 to T2 (*p* = 0.016 and *p* = 0.003). In G2, there was a significant variation only from T0 for T1 in the left ear for an SNR of 10 dB (*p* = 0.001). Conclusion: Speech perception in noise significantly improved after auditory training in old adults.

## 1. Introduction

Language has been a major driver for the human brain’s evolution. As social beings, humans need to be endowed with verbal language as it is essential to communicate with coherent speech and to adequately perceive meaning and intention [1].

Audition belongs to a complex system that receives sound vibrations through the tympanic membrane, which in turn are converted into sound signals that, in a final phase, are transmitted through the auditory pathway to the brain. All this processing is a key part for sound perception [1,2].

In order to get a correct perception of sounds, the subject’s auditory system must be intact. This system is subdivided in two parts: the peripheral part, to which the outer, middle and inner ear belong, and the central part that goes from the first synapse, with its origin in the cochlea, and spreads to the auditory cortex [2,3].

Thus, auditory information covers the peripheral auditory system and the auditory pathway up to the cortex, allowing the subject to detect, discriminate, locate, identify and recognize sound in silence and noise, and finally interpret it [4].

Studying the auditory system comprises not only checking if a person hears properly, with calibrated equipment and specific tests such as the Simple Tonal Audiogram and the Vocal Audiogram, but also checking if the person can process and interpret the sound properly [5,6].

The importance of evaluating auditory processing derives from the acknowledgement that most adults present changes at the central auditory processing (CAP) level due to ageing or to a subjacent neurologic involvement [7,8]. Checking for CAP involves behavioral auditory tests in controlled ambiances in which the participant is evaluated in terms of how the sound is analyzed and how the acoustically conveyed information is interpreted [3,8].

CAP is defined as the set of auditory system mechanisms and processes that is responsible for several behavioral tasks such as sound location and lateralization, auditory discrimination, recognition of sound features (intensity, duration and sound frequency), temporal aspects and auditory performance in the presence of competitive noises, making it possible to carry out diagnosis through verbal and nonverbal sounds [6,9,10]. This can be evaluated through different tests that study the various functions of the auditory central nervous system (ACNS) such as binaural interaction tests, monaural low redundancy speech tests, temporal processing tests and dichotic speech tests [11,12,13,14,15,16]. There are differences between hemispheres in the normal processing of speech sounds presented in a dichotic way, where the right ear has an advantage over the left ear in terms of speech sounds. This phenomenon happens because auditory speech stimuli captured by the right ear are directly processed in the left hemisphere (responsible for speech processing) through the action of the contralateral pathways. When speech stimuli are captured by the left ear, they are first directed to the right hemisphere and are later sent via the corpus callosum for processing in the left hemisphere [15,17]. Monotic tests, such as the speech in noise test, or a filtered speech test, activate the auditory system’s ipsilateral and contralateral pathways. This mechanism neutralizes the effect of laterality and promotes a similar performance between the ears [18].

In this study, our main focus was on monaural low redundancy speech tests, in particular the speech in noise test and the filtered speech test, that assess auditory closure. A words-in-noise task adds significant cognitive load, compared to a similar task without noise. However, if the word recognition score in a quiet environment is poor, it will generally indicate that performance in noise will be poorer [19].

According to the American Speech-Language-Hearing Association (ASHA) [20], CAP disorder impairs at least one of the tasks described previously. Certain patients present with complaints regarding the discrimination of speech in noisy environments, the memorization of words or numbers or even understanding and complying with instructions, notwithstanding having auditory thresholds within normal range and fairly preserved cognition. This implies that, even though the peripheral auditory system is conserved, central auditory processing is sometimes impaired compromising the analysis of auditory information. Thus, CAP diagnosis must also include the evaluation of auditory thresholds [5,20,21].

In current our society it is estimated that almost 12% of individuals who are norm hearers have difficulties in the perception and discrimination of sound stimuli, and the population under 50 years reveals an approximate deficit of 37% in the discrimination of sound stimuli when they face a competitive noise [22,23].

Age-dependent changes imply a wide range of functional declines in the old adult that spans psychological, physiological and biological aspects. Through these senescent processes, the appearance of bilateral progressive sensorineural auditory changes is common, as is the case of presbycusis caused by degeneration of the ciliated cells of the inner ear. Therefore, it is correct to assert that the old adult will have difficulties in the discrimination of speech as all the changes mentioned above will have an impact on CAP level [24,25,26] and, consequently, impair overall quality of life at a psychosocial level [25,26,27].

Previous research, such as that published earlier by Bocca and collaborator [28], revealed the presence of a high percentage of errors in filtrated speech tests, confirming the difficulty of the old adult to understand and process speech due to the ageing process and the consequential changes at the CAP level [7].

Ageing also has negative effects on the corpus callosum through a slow and progressive deterioration that damages the interhemispheric links and further contributes to limit the understanding of speech [7,29,30].

Apart from changes at the corpus callosum level, other age-dependent changes along the auditory pathway, particularly at the peripheral zone and its intertwining with the ACNS, contribute to a fragmentation of language perception and understanding [29,30].

Results from auditory processing disorder (APD) diagnostic tests are pivotal to the understanding of the main individual difficulties and to allow for a tailored auditory training approach, taking into account the subject’s limitations and focusing mainly on the areas that must be stimulated [31,32,33,34]. This type of training can be divided into two types: formal training that is carried out with the presence of an audiologist with acoustically calibrated equipment in an environment that can be controlled acoustically, and informal training that can be carried out by the individual at home through games that permit stimulation of auditory skills. Both types of training consist of the repetition of several tasks adding a higher degree of difficulty after conclusion of each stage [6,34].

Auditory training is constituted by a set of exercises and strategies that permit improvement of auditory skills in which the subject has difficulty [6,34]. According to some authors, formal auditory training must have between 8 and 12 sessions at least once a week, whereas informal training must be done at home 20 min daily [34].

According to Pereira’s study in 1993, the environment, and all the hearing requirements with which the subject come across in their daily life, play a fundamental role in the conservation and reinforcement of the auditory skills trained during the auditory training sessions [35,36,37]. Therefore, the aim of this study was to evaluate CAP before and after auditory training in the old adult with hearing loss, addressing the acute and sub-acute (after three months) therapeutic benefits, after three months, of a tailored intervention over CAP.

## 2. Methods

### 2.1. Design, Population and Procedure

A nonrandomized intervention study was implemented enrolling 15 old adults (9 females) with an average age of 78.6 ± 10.9 years with mild to moderate hearing loss and low literacy. All were members of a day care center in Coimbra district, Portugal.

All the individuals who participated in the study had a normal bilateral otoscopy, bilateral type A tympanogram and a medium tonal loss under 50 dB in both ears. Only one of all the participants benefitted from two hearing aids. An obliterating presence of cerumen, type B tympanogram, presence of neurological diseases and mean tonal loss greater than 50 dBHL were considered as exclusion criteria. This last criterion was used because the stimuli were presented at 50 dB above average pure tone threshold.

The tests used for baseline screening and selection were otoscopy, tympanogram and a simple tonal audiogram in the frequencies 500, 1000, 2000 and 4000 Hz. Posteriori, all the subjects underwent a speech in noise test with an SNR of 10 and 15 dB. For the tests, a Heine mini 2000 otoscope, a GSI 38 Auto-Timp impedancimeter, a Madsen Midimate 622 Audiometer, TDH 39 headphones, a Samsung computer and European Portuguese Auditory Assessment Battery (BAPA-PE) software (Widex - Aural Rehabilitation, Lda, Portugal) were used [14]. This is the only normalized software adapted for European Portuguese.

In the speech in noise test, 30 dissyllables from a phonetically balanced list were used, with speech in ipsilateral presentation and competitive noise, with a signal/noise ratio of + 10 and + 15 dB. The speech (signal) was always presented at the same intensity—50 dB above the average hearing thresholds measured previously with a simple tonal audiogram while the noise was presented 10 dB or 15 dB below the signal. The noise used in the test was a recording of conversations simulating the daily life ecosystem of the participants. This type of noise covered the entire spectrum of the word. The test was started by the right ear and random presentation of the stimuli used was managed by the BAPA-PE software. During the test, the subject was asked to repeat each word heard. There was no feedback given to the subjects to avoid learning bias. The reference criterion for the ability of auditory closure in this test was set as a success rate above 70% for both ears [15].

In the filtered speech test, the frequencies of speech sounds were filtered in order to simulate unintelligible or poorly understood speech. When an individual has normal central auditory processing, he is able to perform auditory closure, filling in distorted or missing parts of the auditory signal and recognizing the message. In the filtered speech test, 40 words of consonant-core-consonant (CNC) were used in each ear through a low-pass filter of 500 Hz and a rejection rate of 18 dB per octave. The stimuli were displayed unilaterally 50 dB above the average hearing thresholds measured previously with a simple tonal audiogram. The test was started through the right ear and the stimuli were randomly presented in all experimental conditions. Stimuli randomization was managed with the BAPA-PE software. During the test, the subject was asked to repeat each word heard. There was no feedback given to the subjects to avoid learning bias. The normality criterion for this test corresponded to values equal to or greater than 78% of correct answers, with performance of the second ear tested, as a rule, superior to the first [12]. In the study by Martins (2017) the normality cut-off obtained for the Portuguese population was a percentage of correct answers higher than 77% [14].

Cognitive function was evaluated with the Cambridge Neuropsychological Test Automated Battery (CANTAB—Cambridge Cognition, Cambridge, UK) platform [38,39], a language-independent battery allowing for the evaluation of different dimensions of cognitive performance such as memory, sensory motor skills and learning. The selected tests were administered via a tablet, through touchscreen interaction, after a short explanation of the testing procedures. The device was controlled using the CANTAB software (Cambridge Cognition, Cambridge, UK). All study participants were assessed individually. After a first touchscreen adaptation period, participants performed four CANTAB tasks always in the same order: motor screening task (MOT 2.0), spatial working memory (SWM—Recommended Standard 2.0 Extended Tone), paired associates learning (PAL—Recommended Standard Extended Tone) and reaction time (RTI—Simple and Five choice Tone). Due to the long duration required to complete the battery of tests, participants were required to complete the MOT, SWM and PAL, and encouraged to complete the RTI test. The battery was repeated in two moments of evaluation: baseline and after the three-month intervention program. A detailed description of these tasks is provided below, and can be found on the Cambridge Cognition’s website: http://www.cambridgecognition.com/clinicaltrials/cantabsolutions/tests.

The MOT test consisted of colored crosses appearing randomly in different locations on the screen, one at a time. The participant was asked to select the cross on the screen as quickly and accurately as possible. This test provided a measure of sensorimotor deficit or reduced understanding of simple tasks. The obtained variables were motor latency (in ms) and total accuracy (maximum 10 points).

PAL is a visuospatial associative learning task addressing visual memory and new learning skills, in an object-location paradigm. In this test, six boxes were displayed on the screen and were opened in random order to reveal the content (a specific pattern). The participant memorized the pattern of each box and, afterwards, matched each individual pattern to its corresponding box. No time limit was enforced during the task. The number of patterns presented increased if participants were able to correctly locate the original location of every pattern on the first attempt, up to a maximum of eight patterns (stage 5). If the participant failed to recall the locations correctly, the patterns in that stage were represented for up to 10 attempts per stage, after which the task was terminated. The variables extracted for analysis were memory score (higher score indicates better associative learning), achieved level and total adjusted errors.

SWM is a self-ordered searching task requiring participants to maintain and update spatial information in working memory, evaluating the ability of the individual to manipulate stored information. It’s also a useful tool to assess the heuristic strategy of participants. The task involved selecting boxes and using a process of elimination, by which the participant needed to find one yellow token in each of a number of boxes, and use them to complete an empty column on the right-hand side of the screen. Depending on the difficulty level used for this test, the number of boxes increased progressively. The participant needed to make multiple searches within one trial. Spatial working memory was indexed to the between-search error, which was scored whenever a participant returned to a box where a token was already found. Additionally, the task provided a measure of heuristic searching strategy, which was scored from 1 to 37, with higher scores indicating less evidence of strategy use. Other measures included the achieved level and the total errors.

After carrying out peripheral and central evaluation of the auditory system, formal auditory training started. Formal auditory training constituted 10 sessions over 5 weeks where the subjects were divided in two groups. Group 1 (G1), constituted by 7 individuals, underwent auditory training based on the speech in noise test and group 2 (G2), constituted by 8 individuals, underwent the filtered speech test. This distribution was made according to the difficulties showed during the evaluation of the speech in noise test. Those who revealed more difficulties in the speech in noise test were integrated into G2. The difficulties were not associated with any type of trauma or neurological disease. Both groups had similar clinical and sociodemographic characteristics. G2 presented with a higher value only in medium tonal hearing loss (MTHL) and in the speech-in-noise test.

During auditory training, in all the sessions, sound stimuli were presented randomly in order to prevent order effects as well as learning and familiarity effects. Informal training at home was not implemented, hence a twice a week schedule was adopted for the formal intervention plans. An audiologist always performed the formal auditory training and the session lasted approximately 30 min.

After carrying out all the sessions of auditory training, we proceeded with the simple tonal audiogram and with the speech in noise test with the same parameters used in the baseline assessment.

Three months after auditory training, participants were re-evaluated with the speech in noise test with a SNR of 10 and 15 dB.

### 2.2. Statistics

Statistical analysis was performed using software IBM SPSS^®^ 24.0 (National Opinion Research Center, Chicago, IL, USA) and for graphical representations the GraphPad Prism 6.04 (La Jolla, San Diego, CA, USA) was used.

All continuous results were expressed as a mean ± standard deviation (SD) and categorical variables were reported as frequencies and percentages. To evaluate the normality of distribution of continuous variables, the Shapiro-Wilk test was used. To compare data between G1 and G2, the nonparametric U Mann-Whitney test was used. To compare data between the baseline and the postintervention evaluation, the paired Student’s *t*-test was used when distribution was normal, otherwise the Wilcoxon test was used. Fisher exact tests were used to compare categorical variables. A 2-factor mixed-design ANOVA was used to evaluate the average percentage of correct answers in both ears and for the SNRs considered (10 and 15 dB). Evaluations were made at three different times: before auditory training (T0), immediately after auditory training (T1) and 3 months after auditory training (T2). The Greenhouse-Geisser correction was used when sphericity was violated, and the Bonferroni adjustment was adopted for multiple comparisons designed to locate the significant effects of a factor.

Differences were considered statistically significant at *p*-value of less than 0.05.

### 2.3. Ethics

The study was conducted according to the guidelines of the Declaration of Helsinki and approved by the Ethics Committee of the Polytechnic Institute of Coimbra (Process number 8/2018). The anonymity and confidentiality of the collected data were guaranteed and all participants signed an informed consent prior to the study. There was no conflict of interest declared.

## 3. Results

Both groups had similar clinical and sociodemographic characteristics. Significant differences were observed only in medium tonal hearing loss (MTHL) and in the speech-in-noise tests. Although medium tonal hearing loss (MTHL) was higher in G2, the classification of hearing loss was the same in both groups and corresponded to mild hearing loss with an average tone loss between 21 and 40 dB. There were no differences between groups in the assessed cognitive dimensions. (Table 1).

At baseline, the hearing thresholds for the right and left ears were 26.0 (±8.9) and 22.3 dB (±8.4), respectively, obtained at 500 Hz, and of 68.7 (±20.4) and 64.7 dB (±22.9), respectively at 8000 Hz. After auditory training, the thresholds decreased and an improvement in hearing thresholds from 250 Hz to 500 Hz was observed in both ears, even though the criterion for statistical significance was not reached (*p* ≥ 0.05). As depicted in Figure 1, similar findings were observed regarding hearing thresholds at different frequencies according to the groups G1 and G2.

Figure 2 depicts the success rate in the speech in noise test for the right and left ears in all the participants. The performance of the individuals in the speech in noise test was very low, where we verified that no individual reached the reference criterion for the ability of auditory closure previously defined in this test, that is, a success rate above 70% for both ears [15]. As can be observed in Figure 2, the success rates in T0 for an SNR 10 and 15 dB, in the right ear, were 22.6% (±15.4%) and 32.5% (±18.7%), respectively. In the left ear, the success rates were 29.4% (±19.6%) and 32.8% (±21.2%), respectively. As represented, there was a statistically significant increase in the success rate from baseline (T0) to the evaluation immediately after the training sessions (T1), in both ears, and this effect persisted in the evaluation three months after the training sessions (T2). However, the reference criterion for the ability of auditory closure of this test was not reached even after auditory training. No significant differences were found in the comparison between T1 and T2 for both ears (Figure 2).

Figure 3 depicts the success rate for the right and left ears in G1 (upper panel) and G2 (bottom panel). Statistically significant changes in the success rates were identified mostly in G1, with an increase from T0 to T1 and the maintenance of the mean success rates from T1 to T2 in both ears. In G2 there was a significant variation only from T0 for T1 in the left ear for an SNR of 10 dB, even though an overall trend for improvement in the success rates was also observable.

Factorial analysis showed that training had a significant effect on success (F (2) = 22.221; *p* < 0.001), with an absence of significant effects either for the ear factor (F = 1.142; *p* = 0.327) and for the age factor (F = 0.359; *p* = 0.706). In other words, success was not influenced by the ear or by the age of the participants. No significant interaction terms were identified in the factorial analysis (F = 0.241; *p* = 0.787). An overall increase in success was observed in all populations, with the composite success rate increasing from 29.3% (±18.1%) at T0, to 40.7% (±25.4%) at T1 and reaching 41.7% (±24.8%) at T2 (see Figure 4). These results highlight the effect of the training sessions (increase from T0 to T1) and the maintenance of the benefit over a period of 3 months (no change from T1 to T2).

## 4. Discussion

Hearing loss is a common feature in ageing, having important consequences on speaking and comprehension in noisy environments, and thus affecting communication in the old adult. Considering the results obtained in the simple tonal audiogram and the average of medium hearing tonal loss at baseline in our study, a mild degree of hearing loss for both ears was identified according to the standard 02 of the Bureau International d’ Audiophonologie (BIAP) [40]. These values showed small improvement after auditory training with no statistically significant differences in medium tonal loss when we compared the conditions before and after auditory training either for the right ear (*p* = 0.864) or for the left ear (*p* = 0.838). These results are in accordance with the Musiek & Baran study [4], in which auditory training also had no influence at the level of hearing thresholds and medium tonal loss. In spite of the mild degree of hearing loss in our participants, a low percentage of success in the speech in noise test was found for both ears, irrespective of the SNRs (10 or 15 dB). The average success rates were 22.6% (±15.4%) for an SNR of 10 dB and 32.5% (±18.7%) for an SNR of 15 dB, regarding the right ear, and 29.4% (±19.6%) for an SNR of 10 dB 32.8% (±21.22%) for an SNR of 15 dB in the left ear. These results indicate APD, which translates into an increased difficulty in understanding speech in competing noise situations, even in the presence of a small hearing loss.

Research and clinical practice concerning ageing and auditory communication have been driven by questions about age-related differences in peripheral hearing, central auditory processing and cognitive processing. Pichora-Fuller & Gurjit Singht [41] proposed an integrated framework for understanding how auditory and cognitive processing interact when old adults listen, comprehend, and communicate in realistic situations, to review relevant models and findings, and to suggest how new knowledge about age-related changes in audition and cognition may influence future developments in hearing aid fitting and audiological rehabilitation. In this study, the evidence suggested that auditory difficulties cascade upwards, such that higher-level cognitive processing involving memory is compromised because mental resources are reallocated to perception and away from storage. When effort is focused on word identification, mental resources for storing heard information and for constructing meaning from ongoing discourse are depleted. Accordingly, any intervention, including hearing aid fitting and auditory training that improves auditory processing to make word identification less effortful, should free resources for higher-level processing, with a consequent improvement in cognitive measures such as working-memory span.

After undergoing ten sessions of auditory training, we observed statistically significant differences in understanding speech. We also observed that auditory training with the speech in noise test used in G1 was particularly efficient regardless of SNR. Conversely, the efficacy of the training protocol in G2 was not equally efficient. Ferguson & Henshaw [42] concluded that auditory training results in generalized improvements in measurements of self-reported hearing, competing speech, and complex cognitive tasks that all index executive functions. This suggests that for auditory training related benefits, the development of complex cognitive skills may be more important than the refinement of sensory processing. Furthermore, outcome measures should be sensitive to the functional benefits of auditory training. Similar to the study of Anderson et al. [43], we verified that, after training, the group of individuals with hearing loss showed a reduction in neural representation of the speech envelope presented in noise, approaching levels detected in old adults with normal hearing. Importantly, changes in speech processing were accompanied by improvements in speech perception. Thus, central processing deficits associated with hearing loss may be partially remediated with training resulting in real-life benefits for everyday communication.

The results obtained in the speech in noise test after the auditory training (T1) and 3 months after auditory training (T2), and regardless of SNR, showed no statistically significant differences, implying that all the skills acquired and improved during the training sessions were maintained after 3 months. Also, the auditory training undergone by the G1 participants was efficient. In G2 the auditory training contributed to a trend of improvement although not reaching statistical significance in all conditions. This could be explained by the fact that G2 had more difficulty in the speech in noise test performance and a need to personalize the training protocol to include the filtered speech test in order to make auditory training easier and more motivating. Thus, it is possible that the efficiency of training in G2 was determined either by the type of training, the characteristics of the group or a combination of both factors.

Several previous research studies also demonstrated the benefits of auditory training in old adults, demonstrating its association with significant improvements in the recognition of sounds, words and sentences, even in competitive noise situations [44]. It is possible with just a few sessions of auditory training to improve the discrimination of speech in noise even in subjects with hearing loss [8,42,45].

## 5. Limitations

This study had limitations that should be considered, among which we highlight the small number of participants, which is a significant aspect, although the results are consistent for the most important endpoints considered in the analysis. This was a pilot and preliminary study, without a control group in its design which could have provided added evidence regarding the proposed objective. Another limitation was the fact that there are few monaural low redundancy speech tests in European Portuguese. Despite the randomization of stimuli, and the fact that no performance feedback was given to the participants to control learning bias, we cannot safely say that they did not occur. Also, a longer follow-up would add relevant information concerning the long-term benefits of the intervention program.

## 6. Conclusions

In view of the strong, positive and statistically significant relation, we can conclude that auditory training positively modulates the performance of CAP evaluation tests. Although auditory training had no influence on MTHL level, it contributed to improve significantly the discriminability and interpretation of sounds in difficult hearing environments. It should be highlighted that according to some studies [35,36,37], and according to the results obtained in our research, the environment and the hearing requirements with which the subject is confronted in his daily life have a key role in the preservation, and in the reinforcement, of the hearing abilities reinforced during auditory training sessions.

Considering all the available evidence, we must acknowledge as a pivotal need the implementation of audiological protocols in old adult populations, including central and peripheral evaluation to orient the old adult to tailored aural rehabilitation programs including auditory training sessions. The whole process will warrant the success of aural rehabilitation programs contributing to enhanced communication abilities and to significant improvement in overall quality of life. This was a pilot study using a methodology that has not entirely yet been implemented in Portugal so the results obtained reflect an initial contribution that we hope will have consequences, not only in the design of other more robust studies, but also in clinical practice with this particular population. The availability of further European Portuguese stimuli is a challenge that needs to be tackled in future research regarding the evaluation of central auditory processing and auditory training.

## Figures and Tables

**Figure 1 ijerph-17-04944-f001:**
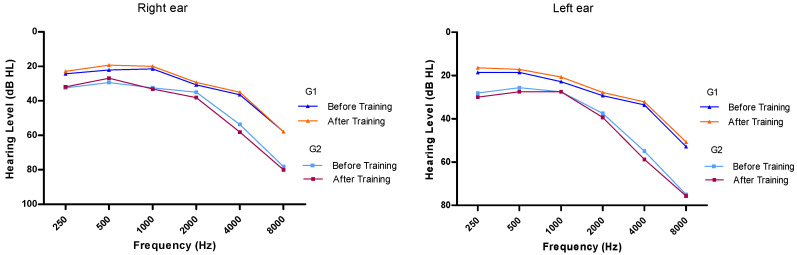
Average threshold obtained by frequency before and after auditory training in the right and left ears in group 1 (G1) and group 2 (G2).

**Figure 2 ijerph-17-04944-f002:**
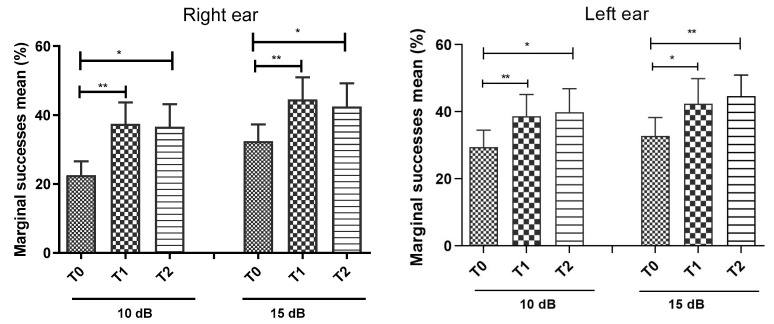
Representation of the marginal success means obtained in T0, T1 and T2 for an SNR of 10 and 15 dB in right ear (left panel) and left ear (right panel). * *p* < 0.05; ** *p* < 0.01.

**Figure 3 ijerph-17-04944-f003:**
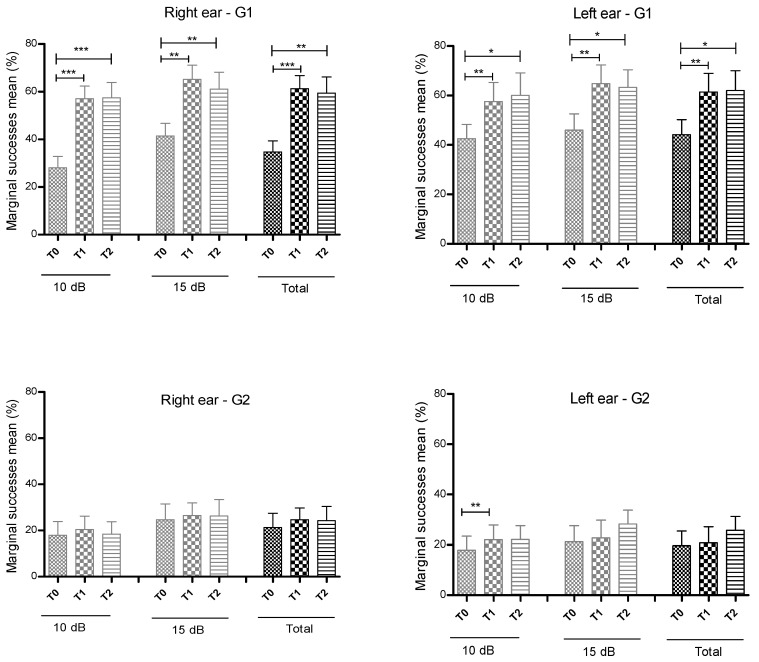
Representation of marginal successes mean obtained at T0, T1 and T2 for a signal to noise ratio of 10 dB, 15 dB and total in group 1 (G1—upper panel) and group 2 (G2—bottom panel) for the right ear (left row) and left ear (right row). * *p* < 0.05; ** *p* < 0.01; *** *p* < 0.001.

**Figure 4 ijerph-17-04944-f004:**
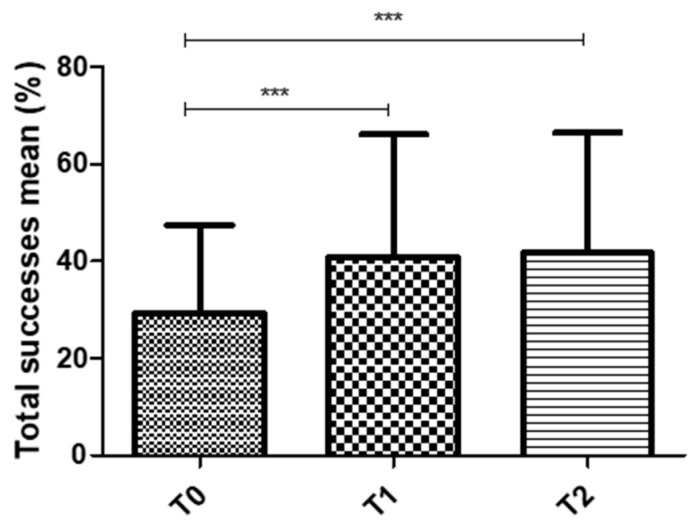
Representation of the composite success (mean of both ears) for overall participants obtained at T0, T1 and T2. *** *p* < 0.001.

**Table 1 ijerph-17-04944-t001:** Demographic and clinical characterization of the G1 and G2 groups presented in mean ± standard deviation.

Variable	G1 (*n* = 7)	G2 (*n* = 8)	*p*-Value
**Gender**			
Female % (*n*) Male % (*n*)	71.4 (5) 28.6 (2)	50 (4) 50 (4)	n.s
**Age. years**	73.6 ± 12.3	83.0 ± 7.8	n.s
**Cognitive Function**			
MOTML	1168.7 ± 340.7	1357.1 ± 380.7	n.s
SWMBE	22.4 ± 5.1	23.9 ± 5.6	n.s
SWMTE	22.4 ± 5.1	24.5 ± 5.8	n.s
SWMS	10.4 ± 0.8	10.3 ± 1.5	n.s
PALFAMS	3.7 ± 1.8	2.8 ± 2.3	n.s
PALTEA	58.4 ± 7.8	59.5 ± 7.9	n.s
PALNPR	4.0 ± 1.6	4.3 ± 1.3	n.s
MTHL **^1^**			
Right ear. dB	27.7 ± 10.1	37.7 ± 9.0	0.04
Left ear. dB	26.1 ± 9.2	36.4 ± 6.7	0.021
**Speech in noise test ^2^**			
Right ear			
SNR 10 dB. %	28.0 ± 12.7	17.9 ± 16.8	n.s
SNR 15 dB. %	41.4 ± 14.0	24.6 ± 19.4	n.s
Total. %	34.7 ± 12.4	21.3 ± 17.5	n.s
Left ear			
SNR 10 dB. %	42.6 ± 15.2	17.9 ± 15.7	0.029
SNR 15 dB. %	46.0 ± 17.3	21.3 ± 17.8	0.029
Total. %	44.1 ± 16.1	19.6 ± 16.4	0.021

^1^ BIAP Recommendation 02/1: Normal or subnormal hearing: the MTHL is below 20 dB; mild hearing loss: the MTHL between 21 and 40 dB; ^2^ reference value for normality: success rate above 70% for both ears [15]; n.s—non statistically significant (*p* > 0.05); MTHL—medium tonal hearing loss; SNR —signal-noise ratio; MOTML—motor latency; SWMBE—mean between-search errors; SWMTE—mean total errors; SWMMS—mean searching strategy; PALFAMS—mean memory score; PALTEA—total adjusted errors; PALNPR—mean level achieved.

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
