# Peer review of "Study of Acute and Sub-Acute Effects of Auditory Training on the Central Auditory Processing in Older Adults with Hearing Loss—A Pilot Study"

_ijerph, 2020, doi:10.3390/ijerph17144944_

Round 1
Reviewer 1 Report
First of all, thank you for the opportunity to review this article.
The academic structure is adequate. The different sections allow a pleasant and easy reading. An important value of the article is its explanatory simplicity, following the principle of parsimony. I want to congratulate the authors on this way of presenting their study.
The methodology is correct, and the instruments are adequate. Statistical tests distinguish between normal and non-normal distributions and use parametric and non-parametric tests.
However, with a small sample, would it not have been better to always use nonparametric tests? What do you think about it? Surely the results would not change, but have they checked?
I would like to make the following considerations, in order to help, if possible, to improve the manuscript:
- Why does the sample show so much dispersion in age, with a small n? I believe that the authors should explain this dispersion and whether or not it induces the results. Could the variable age be a control variable?
- Lines 58-60 talks about: “The importance of evaluating the auditory processing derives from the recognition that most adults present changes at the central auditory processing (CAP) level due to ageing or to a subjacent neurologic involvement”. Was this distinction (aging versus injury) taken into account in selecting groups? Shouldn't it be a control variable?
- Lines 143-144: “This distribution was made according to the difficulties showed during the evaluation of the speech in noise test”. Are these difficulties due to some type of trauma, chronic disease ..., or are they due to physical deterioration caused by old age?
I reiterate my congratulations for the work.
Author Response
Dear reviewer,
We are grateful for the availability in revising our article, as well as all the considerations, suggestions and recommendations proposed. We are sure that it will be an important contribution for the improvement of this work.
We will then answer each question point by point.
Best Regards,
Carla Matos Silva

Reviewer 2 Report
This is a generally well described and well executed study of patients with presbycusis and central auditory processing disorder. The initial results showing improvement in processing of auditory information after training were confirmed by the follow up testing. The main flaw, as acknowledged by the authors, is the small sample size. They could add "a pilot study" to the title or could describe their research as a pilot study in the abstract.
Author Response
Dear reviewer,
We are grateful for the availability in revising our article, as well as all the considerations, suggestions and recommendations proposed. We are sure that it will be an important contribution for the improvement of this work.
Best Regards,
Carla Matos Silva

Reviewer 3 Report
Summary
The authors gave auditory training to 15 older adults with hearing losses, consisting of 10 training sessions over 5 weeks. Participants were divided into two groups that either were given auditory training based on a speech-in-noise test, or a filtered-speech test. The group trained using the speech-in-noise test showed improved performance following training, but the group trained using a filtered-speech test did not.
General comments
I have a number of reservations regarding the manuscript. The Introduction is rather too general in its current form, and is rather unclear regarding what the gap in the literature is, what the hypotheses are, or what is novel about the study. It is not clear from the text in the Introduction whether previous studies that discuss or test auditory training have addressed CAP level. No feedback was given and the training seems to have consisted of repetitions of the same set of stimuli within the tests, so it is not clear whether the authors’ conclusion that “speech perception in noise significantly improves after auditory training in old adults” is valid, or whether the participants simply learned the stimuli over a number of sessions. The manuscript needs to be clearer regarding the procedures and methods, which are described too briefly. Some of the terms are used erroneously (e.g. frequency appears to be confused with SNR in a number of places), and one of the tests (the filtered-speech test) is not described at all. The standard of English needs some improvement, as it is currently rather difficult to follow the manuscript (e.g. “Those who revealed more difficulties in the speech in noise test integrated the G2”). The criteria for the division of the participants into two groups is not described adequately. It seems odd that the authors mention in the title that the study involves “Long-Term Benefits,” as the followup testing only took place 3 months after the training, and the authors themselves acknowledge this limitation in their Discussion. In some places statements are made that are not supported by data (see below). I am not sure how these issues can be resolved without a major overhaul of the manuscript. My recommendation is therefore reject. I have provided further comments below, that I hope will be useful to the authors.
Specific comments
Line 23: “enrolling 15 old adults…”
Suggest “enrolling 15 older adults”
It is stated on line 117: “Therefore, the aim of this study was to evaluate the CAP before, after and 3 months after auditory training in older people with hearing loss, addressing the acute and long-term therapeutic benefits of a tailored intervention over the CAP.”
The text here (and elsewhere, including the title regarding “Long-Term Benefits”) should be rewritten, as CAP assessment only 3 months after the intervention does not allow conclusions to be drawn regarding long term benefits – a much greater time course of follow up study would be required for this.
Lines 68-70 mention that the mechanisms and processes underlying the CAP “can be evaluated through different tests that study the various functions of the Auditory Central Nervous System (ACNS) such as the binaural interaction tests, low redundancy monaural speech tests, temporal processing tests and finally the dichotic speech tests [11-15].” Near the end of the Introduction on line 117, the text makes it look as if the CAP will be fully evaluated, and the reader might suppose that all of the tests mentioned on lines 68-70 will be tested. It only becomes clear later on that the study consists of speech in noise tests and filtered speech tests. This needs to be clearer, and the hypotheses clarified.
Line 121: Design, population and procedure
This section is too short and needs more detail. For example, line 140 states that “The formal auditory training was constituted of 10 sessions during 5 weeks where the subjects were divided in two groups.” On what bases were these decisions made? Why was 10 sessions deemed to be sufficient? Why only 5 weeks if the aim of the study was to evaluate the long-term benefits of the intervention? Also, no information is provided regarding what equipment was used, but this needs to be included.
Line 126: “Only one of all the participants benefitted from two hearing aids.”
What about the other participants? Did they have one hearing aid only, or no hearing aids?
Line 137: “The reference criterion for the ability of hearing closure in this test is a success rate above 70% for both ears [14].”
I do not understand this. What is meant by “hearing closure?” What does it mean that, on average, none of the participants achieved this in the Experiment (Figure 3)?
Lines 139-148: The auditory training is described here, but the description is too brief to be clear about what is happening. The “training” appears to be “…10 sessions during 5 weeks where …G1…underwent an auditory training based on the speech in noise test and …G2…underwent the filtered speech test.” So did the training just consist of repeated sessions of the speech in noise test/filtered noise test? It appears there was no feedback given, but if the intention is to put together an intervention that will improve performance, why not give feedback?
The lack of clarity and detail regarding the stimuli and procedure is worrying, as it is not clear if the same sessions were just given over and over, so that the participants simply learned the stimuli set and the appropriate responses. Were novel stimuli provided for each session? Randomising the order of the presented stimuli (line 147) would not address this issue.
Also, what was the duration of each of the sessions? This needs to be reported.
Line 141: “The group 1 (G1), constituted by 7 elements, underwent an auditory training based on the speech in noise test and the group 2 (G2), constituted by 8 elements, underwent the filtered speech test.”
What is meant by “elements” here? Please clarify.
Line 142: “…group 2 (G2), constituted by 8 elements, underwent the filtered speech test.”
What is the filtered speech test? Unless I have missed it, there is no description of it in the manuscript. How is it different from the speech in noise test that group 1 undertook?
Line 144: “Those who revealed more difficulties in the speech in noise test integrated the G2.”
Does integrated here mean that the participants were assigned to Group 2? What criteria were used for this assignment (what defined the participant as “having more difficulty”)?
Line 152: “Three months after the auditory training central auditory processing we re-evaluated, through speech in noise test with a SNR of 10 dB and 15 dB.”
For G2 as well? Should they not have been tested using the filtered speech test?
Line 163: “…for the frequencies considered (10 dB and 15 dB).
But 10 dB and 15 dB are not frequencies, they are SNRs – please correct.
Line 175: “MTHL”
This is the first mention of this term – it needs to be defined.
Lines 175-177: “At baseline, the MTHL for the right and left ears were of 26.0 dB (±8.9dB) and 22.3 dB (±8.4dB) respectively, obtained from 500Hz, and of 68.7dB (±20.4dB and 64.7dB (±22.9dB) respectively, obtained from 8000Hz.”
Are these results reported for all participants? Is so this needs to be reported. Why are the results not reported separately for groups 1 and 2?
Line 184: “Figure 2 depicts the success rate…”
On the figure the y axis reads “marginal successes mean (%)”. Is this the same as the success rate? Suggest using consistent terms.
Line 203, and throughout (e.g. line 224): “the success rates were not influenced by the ear or by the frequency of the noise.”
But frequency was not manipulated in the study. Do the authors mean “SNR”?
Line 223: “an unexpected low percentage of success in the speech in noise test was found…”
What is being used as a comparison here? Have other studies used similar parameters and found different results? If so they should be cited here, otherwise more explanation is needed as to why the results were so low.
Line 236: “information degradation hypothesis”
What is this? More detail is needed.
Line 252: “Similarly to the study of Anderson et al. [39], we verified that after training, the auditory training group with hearing loss experienced a reduction in the neural representation of the speech envelope presented in noise, approaching levels observed in normal hearing older adults.”
I do not see how this statement is supported, as I cannot see anywhere in the manuscript where the authors measured a reduction in the neural representation of the speech envelope presented in noise.
Line 262: “However, this could results from the fact that G2 had more difficulty in the speech in noise test performance and the need to personalize the training protocol to include the filtered speech test in order to make the auditory training easier and more motivating. Thus, it is possible that the efficiency of the training in G2 is determined either by the type of training, the characteristics of the group, or even the combination of both factors.”
Was it not simply that the experimental parameters chosen were not appropriate for G2? Why was Hearing Level not reported separately for G1 and G2? If G2 had more severe hearing losses, then the task was probably simply too difficult for them, as suggested by Figure 3 that shows that they were unable to achieve a success rate greater than 30%, in contrast to G1.
Figure 3: The axes should be the same for the upper and lower panels to make comparisons easier. At present, the upper panels are cut off at 80% compared to only 40% for the lower panels.
Author Response
Dear reviewer,
We are grateful for the availability in revising our article, as well as all the considerations, suggestions and recommendations proposed. We are sure that it will be an important contribution for the improvement of this work.
We took into account your recommendations, which we are very grateful for, so we changed some parts of the article, namely the title, the objective, the introduction and the methodology for a better explanation of the study
We will try to respond step by step to each of the recommendations you have proposed in order to better clarify the article.
Best Regards,
Carla Matos Silva

Reviewer 4 Report
This article deals with the benefits of auditory training for older adults with hearing loss. The main research findings will be important for understanding their speech perception ability in noise. However, several points as indicated below need to be addressed by authors to improve the quality of the article.
Revision suggestions:
1. In this article, the subjects were divided in two groups. However, authors did not show subject’s group profiles, for example, range of their age, hearing level, speech intelligibility, and so on. It is important information to interpret the difference of result whether the groups was equivalent.
2.It may be helpful for understanding the results to provide more information about speech in noise test. For example, where did you conduct this test? How did you present the test stimuli? How is the number of test trials? Which ear did you present the test syllables to? Did you use the different test syllable list for evaluation before and after auditory training (T0, T1, T2)?
3. The auditory trainings is the same as that. It is needed to provide more information in detail. Otherwise, we are not able to interpret the differences before and after auditory training. Did you use similar stimuli for auditory trainings with speech in noise test? If so, the differences before and after training is interpreted as learning effect. If not so, is it correct to judge the differences as the improvement of central auditory processing?
4. Could you indicate the purpose of analyzing each ear’s result separately? If the authors assume the differences between ears, you had better to show the sufficient cause at introduction and discussion.
5. In results, authors include the contents of discussion (line 179-180). It should be distinguished the fact of result from the speculation of discussion.
6. Additionally, the article would benefit tremendously from language editing by either a native English speaker or a professional editor. There are a number of typographical and grammatical errors throughout the article.
Author Response

(The authors gave the same response as above.)

Reviewer 5 Report
The authors work on a very interesting and important topic.
Unfortunately the scientific design of the study has severe faults (see attached file)
I would recommend additional experiments with more participants and control group.
Review of the Article:
Introduction:
The introduction is well done. The authors mention several tests in order to detect ACNS (68-70) and they describe different diseases (71-78) which may impair ACNS in older people. The aim of the study could be presented more differentiated (see comments in „Results“).
Methods:
- Which tests have been done on the patients? What were the defects in the ACNS? How was cognition, memory-spam etc tested in the patients? In the discussion you mention "normal hearing in older adults" (255). How is it defined and how was the hearing of the patients included in this study?
- How was the selection made in order to divide the patients into the two therapy groups? What are „difficulties“ (144), you should describe them exactly.
- What exactly was the training the Audiologist (filtered speech test) did in group 2? What was the training at home for both groups? What exactly did the patients do during 20 minutes?
- How was done the control of the training at home?
- The small number of participants is a problem, indeed.
Results:
In G1 there was seen a significant improvement (of what exactly? Results of speech tests, cognition tests, memory-spam?), but not in G2.
In my opinion this is all you can say about the results of the experiements. It is not possible to compare both groups, because of different initial reference points concerning the hearing in both groups and because of the different therapies. If you compare it is like comparing apples to pears.
For good interpretation and discussion of your results you need a control group!
With the data presented in this study you can’t prove any efficiency of the training and you can’t explain why you see differences in the two groups. Perhaps the patients of G2 did their training at home less carefully than the patients of G1, or the results show that training has limitations in dependence of the hearing capacities at the beginning of the training (Speech tests?, Cognition tests?).
Author Response

(The authors gave the same response as above.)

Round 2
Reviewer 3 Report
I appreciate the authors efforts made to address the points raised in the last round of reviews. The manuscript is much improved.
I do feel that what needs to be clarified regards how the results relate to the statement in the Introduction that for the speech in noise test, “The reference criterion for the ability of auditory closure in this test is a success rate above 70% for both ears [15],” Line 164, and for the filtered speech test, “The normality criterion for this test corresponds to values equal to or greater than 78% of correct answers, with the performance of the second ear tested, as a rule, superior to the first [12].” Line 175. Do the participants reach these criteria? It does not appear so from looking at the results, so are the results meaningful? This needs clarification.
Author Response
Dear reviewer,
We are grateful for the availability in revising our article, as well as all the considerations, suggestions and recommendations proposed in this round 2. We are sure that it will be an important contribution for the improvement of this work.
We will then answer your question.
Comments and Suggestions for Authors
I appreciate the authors efforts made to address the points raised in the last round of reviews. The manuscript is much improved.
I do feel that what needs to be clarified regards how the results relate to the statement in the Introduction that for the speech in noise test, “The reference criterion for the ability of auditory closure in this test is a success rate above 70% for both ears [15],” Line 164, and for the filtered speech test, “The normality criterion for this test corresponds to values equal to or greater than 78% of correct answers, with the performance of the second ear tested, as a rule, superior to the first [12].” Line 175. Do the participants reach these criteria? It does not appear so from looking at the results, so are the results meaningful? This needs clarification.
Answer: Thank you very much for the suggestion. It is a very relevant aspect that we have taken into consideration. In this sense, we improve the text in the results. (Please see lines 317 – 327)
We clarify that the English of the manuscript has been revised again (Please see yellow underlined)
Reviewer 4 Report
Thank you for your article’s revision.
I could understand importance for Portuguese to show auditory training effects in older adults with difficulties to percept speech under noise condition. The additional information was added and the article was modified. However, there are many points to be improved.
1) Grammatical errors and inappropriate words
There are a number of grammatical errors and inappropriate words. And the present and past tense are mixed up in methods. The article is required extensive revisions to improve language, readability, flow and transition, again.
Some examples are shown below.
line 2 study of the acute … Is it appropriate explanation?
line 81-84 I don’t understand this sentence.
line 145- “ the sound stimulus is presented at 50dBHL above the tonal average of loss”
… Does this sentence mean “the stimuli are presented at 50dB above average pure tone threshold” ?
line 155 In this sentence, the word “with” is over-used.
line 160-162 and 172-173 Authors use same sentence.
Line 168 “40 words of the type consonant-core-consonant” … 40 words of consonant-core-consonant”
Line 171 “50 dB SL above the value of medium tonal loss”
… I don’t understand this explanation. Does authors means “50dB above the average hearing thresholds”?
……….
2) The problem of subject’s group selection
Authors wrote “This distribution was made according to the difficulties showed during the evaluation of the speech in noise test. Those who revealed more difficulties in the speech in noise test integrated the G2. (line 224-)”. Group 1 had good speech in noise scores and were applied to auditory training using speech in noise test, and Group 2 had poor speech scores under noise condition and were applied to the training using filtered speech. However, authors compared different two groups, and concluded “Group 1 revealed to be quite efficient regardless the SNR … No statical significant changes were depicted in G2(line 30-36). This interpretation is not understood for readers.
3) The problem of assessment and training
According to additional information, I knew that authors used same stimuli in assessment and training. In limitations, authors wrote “the subjects were not given performance feedback, to control learning effects, we cannot safely say that they did not occur (line 422-423).” G1 received 10 sessions training during 5weeks, G1 subjects learned the test stimuli without debut. In case of considering training effects, authors must use different test stimuli with training stimuli. This point is methological mistake.
The grammatical errors and inappropriate words are improved by native English speakers, however, it is difficult to modify the methological problems (2, 3). Therefore, it is needed to examine the study design and method to clarify author’s hypothesis again. I am anticipating your future research.
Author Response
Dear reviewer,
We are grateful for the availability in revising our article, as well as all the considerations, suggestions and recommendations proposed in this round 2. We are sure that it will be an important contribution for the improvement of this work.
We will then answer each question point by point.
Comments and Suggestions for Authors
Thank you for your article’s revision.
I could understand importance for Portuguese to show auditory training effects in older adults with difficulties to percept speech under noise condition. The additional information was added and the article was modified. However, there are many points to be improved.
1) Grammatical errors and inappropriate words
There are a number of grammatical errors and inappropriate words. And the present and past tense are mixed up in methods. The article is required extensive revisions to improve language, readability, flow and transition, again.
Answer: We appreciate your comment and clarify that the English of the manuscript has been revised (Please see yellow underlined)
Some examples are shown below.
line 2 study of the acute … Is it appropriate explanation?
Answer: Thank you very much for your suggestion. This point is corrected (Please see line 2)
line 81-84 I don’t understand this sentence.
Answer: Thank you once again. This point is corrected. (Please see lines 80-81)
line 145- “ the sound stimulus is presented at 50dBHL above the tonal average of loss”
… Does this sentence mean “the stimuli are presented at 50dB above average pure tone threshold” ?
Answer: We change the text according your suggestion. (Please see lines 142 – 143 and 153 - 154)
line 155 In this sentence, the word “with” is over-used.
Answer: We change the text according your suggestion. (Please see line 153)
line 160-162 and 172-173 Authors use same sentence.
Answer: We appreciate your comment and clarify that the phrases refer to two different tests.
Line 168 “40 words of the type consonant-core-consonant” … 40 words of consonant-core-consonant”
Answer: Thanks for the suggestion. We correct the text. (Please see lines 164 and 165).
Line 171 “50 dB SL above the value of medium tonal loss”
… I don’t understand this explanation. Does authors means “50dB above the average hearing thresholds”?
Answer: Thank you very much for the comment. We change the text. (Please see line 166).
2) The problem of subject’s group selection
Authors wrote “This distribution was made according to the difficulties showed during the evaluation of the speech in noise test. Those who revealed more difficulties in the speech in noise test integrated the G2. (line 224-)”. Group 1 had good speech in noise scores and were applied to auditory training using speech in noise test, and Group 2 had poor speech scores under noise condition and were applied to the training using filtered speech. However, authors compared different two groups, and concluded “Group 1 revealed to be quite efficient regardless the SNR … No statical significant changes were depicted in G2(line 30-36). This interpretation is not understood for readers.
Answer: We appreciate your opinion and we clarify the authors' intention was to verify the effect of training on each group and not the comparison between groups as referred to in line 30 to 36.
3) The problem of assessment and training
According to additional information, I knew that authors used same stimuli in assessment and training. In limitations, authors wrote “the subjects were not given performance feedback, to control learning effects, we cannot safely say that they did not occur (line 422-423).” G1 received 10 sessions training during 5weeks, G1 subjects learned the test stimuli without debut. In case of considering training effects, authors must use different test stimuli with training stimuli. This point is methological mistake.
Answer: Please see lines 439-441.
The grammatical errors and inappropriate words are improved by native English speakers, however, it is difficult to modify the methological problems (2, 3). Therefore, it is needed to examine the study design and method to clarify author’s hypothesis again. I am anticipating your future research.
Submission Date
16 April 2020
Reviewer 5 Report
Much better presentation of the study! The methods and results are clearly described, the discussion is focused on the results and limitations are pointed out. Well done!
Author Response
Dear reviewer,
We appreciate the availability to review our article once again. Your contribution to the revision of this manuscript was very important for the improvement and clarification of this work. We are very pleased with all the changes we have made to meet your expectations.
We clarify that the English of the manuscript has been revised again (Please see yellow underlined).
Round 3
Reviewer 5 Report
Much better presentation of the study! The methods and results are clearly described, the discussion is focused on the results and limitations are pointed out. Well done!
Author Response

(The authors gave the same response as above.)
